# Negative Relationship between Post-Treatment Stromal Tumor-Infiltrating Lymphocyte (TIL) and Survival in Triple-Negative Breast Cancer Patients Treated with Dose-Dense Dose-Intense NeoAdjuvant Chemotherapy

**DOI:** 10.3390/cancers14051331

**Published:** 2022-03-04

**Authors:** Sylvie Giacchetti, Lilith Faucheux, Charlotte Gardair, Caroline Cuvier, Anne de Roquancourt, Luca Campedel, David Groheux, Cedric de Bazelaire, Jacqueline Lehmann-Che, Catherine Miquel, Laurence Cahen Doidy, Malika Amellou, Isabelle Madelaine, Fabien Reyal, Laetitia Someil, Hamid Hocini, Christophe Hennequin, Luis Teixeira, Marc Espié, Sylvie Chevret, Vassili Soumelis, Anne-Sophie Hamy

**Affiliations:** 1Breast Disease Unit (Sénopole), AP-HP, Hôpital Saint-Louis, F-75010 Paris, France; caroline.cuvier@aphp.fr (C.C.); luca.campedel@aphp.fr (L.C.); malika.amellou@aphp.fr (M.A.); laetitia.someil@aphp.fr (L.S.); hamid.hocini@aphp.fr (H.H.); luis.teixeira@aphp.fr (L.T.); marc.espie@aphp.fr (M.E.); 2ECSTRRA Team, Statistic and Epidemiologic Research Center, INSERM UMR-1153, Université de Paris, F-75010 Paris, France; lilith.faucheux@inserm.fr (L.F.); sylvie.chevret@u-paris.fr (S.C.); 3INSERM U976, Université de Paris, F-75010 Paris, France; dgroheux@yahoo.fr (D.G.); jacqueline.lehmann-che@aphp.fr (J.L.-C.); vassili.soumelis@aphp.fr (V.S.); 4Department of Anatomopathology, AP-HP, Hôpital Saint-Louis, F-75010 Paris, France; ch.gardair@gmail.com (C.G.); anne.deroquancourt@aphp.fr (A.d.R.); catherine.miquel@aphp.fr (C.M.); 5Department of Nuclear Medicine, AP-HP, Hôpital Saint-Louis, F-75010 Paris, France; 6Department of Radiology, AP-HP, Hôpital Saint-Louis, F-75010 Paris, France; cedric.de-bazelaire@aphp.fr; 7Immunology, Biology and Histocompatibility Laboratory, AP-HP, Hôpital Saint-Louis, F-75010 Paris, France; 8Surgery Department, AP-HP, Hôpital Saint-Louis, F-75010 Paris, France; laurence.cahen-doidy@aphp.fr; 9Department of Pharmacy, AP-HP, Hôpital Saint-Louis, F-75010 Paris, France; isabelle.madelaine@aphp.fr; 10Department of Surgery, Institut Curie, 26 rue d’Ulm, University Paris, F-75005 Paris, France; fabien.reyal@gmail.com; 11Residual Tumor & Response to Treatment Laboratory, RT2Lab, INSERM, U932 Immunity and Cancer, Institut Curie, 26 rue d’Ulm, University Paris, F-75005 Paris, France; hamyannesophie@gmail.com; 12Department of Radiotherapy, AP-HP, Hôpital Saint-Louis, F-75010 Paris, France; christophe.hennequin2@aphp.fr; 13Department of Biostatistics and Medical Information, AP-HP, Hôpital Saint-Louis, F-75010 Paris, France; 14Department of Oncology, Institut Curie St Cloud–35 rue Dailly, St Cloud, F-92210 Paris, France

**Keywords:** triple negative breast cancers, dose dense neoadjuvant chemotherapy, stromal tumor-infiltrating lymphocyte

## Abstract

**Simple Summary:**

Triple-negative breast cancer (TNBC) continues to have a poor prognosis unless a complete pathological response is achieved after neoadjuvant chemotherapy (NAC). The identification of prognostic factors would help to improve care. The evaluation of tumor-infiltrating lymphocyte (TIL) levels is recommended in patients with breast cancer, but little is known about changes in these levels with treatment. We studied a series of 117 TNBCs treated by dose-dense dose-intense (dd-di) NAC, to identify factors associated with pre- and post-NAC stromal TIL levels, and oncological outcomes. Our findings highlight the complexity of TIL level interpretation and the importance of taking NAC regimen into account when interpreting results. We show that patients with persistently high TIL levels after NAC are at higher risk of relapse and death.

**Abstract:**

*Background:* Patients with triple-negative breast cancers (TNBC) have a poor prognosis unless a pathological complete response (pCR) is achieved after neoadjuvant chemotherapy (NAC). Few studies have analyzed changes in TIL levels following dose-dense dose-intense (dd-di) NAC. *Patients and methods*: From 2009 to 2018, 117 patients with TNBC received dd-di NAC at our institution. We aimed to identify factors associated with pre- and post-NAC TIL levels, and oncological outcomes relapse-free survival (RFS), and overall survival (OS). *Results:* Median pre-NAC and post-NAC TIL levels were 15% and 3%, respectively. Change in TIL levels with treatment was significantly correlated with metabolic response (SUV) and pCR. High post-NAC TIL levels were associated with a weak metabolic response after two cycles of NAC, with the presence of residual disease and nodal involvement at NAC completion. In multivariate analyses, high post-NAC TIL levels independently predicted poor RFS and poor OS (HR = 1.4 per 10% increment, 95%CI (1.1; 1.9) *p* = 0.014 and HR = 1.8 per 10% increment 95%CI (1.3–2.3), *p* < 0.0001, respectively). *Conclusion:* Our results suggest that TNBC patients with TIL enrichment after NAC are at higher risk of relapse. These patients are potential candidates for adjuvant treatment, such as immunotherapy, in clinical trials.

## 1. Introduction

Triple-negative breast cancer (TNBC), defined as tumors with no expression of the estrogen receptor (ER) nor progesterone receptor (PR), and with no amplification of the human epidermal growth factor-2 (*HER2*) gene, accounts for about 15–20% of the more than one million breast cancers diagnosed worldwide annually [1,2]. Patients with TNBC have lower breast cancer-specific survival rates and lower overall survival (OS) rates than those with hormone receptor (HR)-positive and/or with *HER2*-overexpressing tumors [3]. Despite this poor prognosis, TNBCs are highly sensitive to chemotherapy, and are therefore often treated with neoadjuvant chemotherapy (NAC). With this strategy, the tumor response can be rapidly assessed, and research platforms can be established to evaluate the factors predictive of a response to treatment.

The role of tumor-infiltrating lymphocytes (TILs) in breast cancer (BC) has been widely studied over the last decade. Many studies have reported associations between high TIL levels at diagnosis and a better response to NAC [4,5,6,7], together with a better prognosis, in both neoadjuvant and adjuvant chemotherapy settings, particularly for TNBC. The clinical significance of post-NAC TIL levels has been less extensively studied, and their prognostic value remains a matter of debate. Several studies have reported TIL enrichment after NAC to be beneficial [8,9,10], whereas others reported an absence of effect or a worsening of the prognosis [6,11,12,13].

^18^FDG-PET/CT is useful for the initial staging of breast cancers, particularly those of stage IIB or higher. In triple-negative breast cancers, the early metabolic response accurately predicts pCR and outcome. Despite its relatively high sensitivity, interim PET/CT is much less widely used in breast cancers than in other similar disease settings, partly because there is no consensus that an early evaluation of response can be used for the direct modification of BC treatment in the neoadjuvant setting [14]. However, the multicenter randomized phase 2 trial AVATAXHER showed that early assessment by ^18^F-FDG PET was useful for optimizing NAC in the HER2-positive BC subgroup [15].

The concepts of dose density and dose intensity are based on cytokinetics modeling and have been investigated in more than 33 trials. An individual patient-level meta-analysis from the Early Breast Cancer Trialists’ Collaborative Group (EBCTCG) combining individual data for 37 298 patients from 26 randomized trials compared 2-weekly versus standard 3-weekly schedules [16]. The authors showed that increasing the dose intensity of adjuvant chemotherapy by shortening the interval between treatment cycles reduced the 10-year risk of recurrence and death from breast cancer without increasing mortality due to other causes. Our team previously compared the efficacy of two different neoadjuvant regimens (a dd-di cyclophosphamide–anthracycline (AC) regimen and a conventional sequential NAC regimen with cyclophosphamide and anthracycline, followed by taxanes (EC-T)) in a cohort of 267 patients [17]. We showed that the intensified chemotherapy regimen was more beneficial to TNBC patients [17].

Few studies to date have investigated TIL dynamics and TIL levels after NAC in TNBC patients treated with dd-di NAC. We report here findings for a cohort of TNBC patients from a single institution treated with dd-di NAC, with an evaluation of the predictive and prognostic values of TIL levels (before and after NAC) in the population with residual disease and the population with a pathological complete response (pCR).

## 2. Materials and Methods

### 2.1. Study Population

From January 2009 to October 2018, all consecutive patients with TNBC identified as candidates for NAC were treated with a dose-dense dose-intense regimen at the Breast Disease Center of Saint Louis University Hospital, Paris. All these patients were included in this analysis. The study was approved by the hospital’s internal ethics board. Written informed consent was not required by the French law.

### 2.2. BC Diagnosis and Tumor Samples

Breast cancer was diagnosed on 14 or 16 G core-needle biopsies. A lymph node biopsy was also performed if axillary lymph nodes were palpable or identified on axillary ultrasound. Estrogen and progesterone receptor status was determined by immunohistochemistry (IHC), with a cutoff of 10% of the cells stained for the receptor defining positivity [18]. HER2 amplification was systematically assessed by IHC, with SISH used as a control for ambiguous cases. The functional status of the p53 gene was assessed by a functional analysis of separated alleles in yeast (FASAY) method evaluating the transactivation activity of p53 on a p53-responsive promoter stably integrated into the yeast genome [19].

### 2.3. BC Treatment

All the patients received a dd-di NAC regimen (Figure 1). Between 2009 and 2015, patients were treated with six cycles of neoadjuvant cyclophosphamide (1200 mg/m^2^ d1) and epirubicin (75 mg/m^2^ d1) every 2 weeks, followed by surgery and three additional cycles of adjuvant docetaxel (100 mg/m^2^) and cyclophosphamide (600 g/m^2^) every 3 weeks after surgery (SIM1).

From 2016 onwards, patients received four cycles of neoadjuvant cyclophosphamide (1200 mg/m^2^ d1) and epirubicin (75 mg/m^2^ d1) every 2 weeks, followed by four cycles of docetaxel (100 mg/m^2^) every 3 weeks or weekly paclitaxel (80 mg/m^2^) for 12 cycles, followed by surgery (SIM2). None of the patients received capecitabine after the completion of NAC in the SIM1 and in the two cohorts.

If no tumor progression was observed after the completion of chemotherapy, the patients underwent breast surgery and axillary dissection or sentinel techniques, according to initial lymph nodes status. All patients received radiotherapy after surgery on the breast and lymph nodes, according to the initial stage of the tumor. If a patient progressed during chemotherapy, the treatment was stopped and a new treatment was proposed to the patient; either surgery either change in NAC regimen depending notably on whether the tumor was deemed operable

### 2.4. ^18^FDG-PET/CT Imaging and Measurement of the Metabolic Response

^18^FDG (5 MBq/kg) was administered and imaging was initiated about 60 min later. The Gemini XL PET/CT scanner (Philips Medical systems) was used. ^18^FDG-PET/CT scans were performed at treatment initiation and after two courses of NAC, to evaluate the early metabolic response (optional). The metabolic response was evaluated by determining the change in maximal standardized uptake value (SUV) in the breast tumor, and was defined as the percent change in SUV modification after two courses of NAC: ΔSUV_max_ (%) = 100 × (2nd cycle SUV_max_ − baseline SUV_max_)/baseline SUV_max,_ as previously described [20].

### 2.5. Pathological Review

Paired pretreatment core needle biopsies and post-NAC surgical specimens were reviewed for the purposes of the study. Two pathologists (AdR and CG) specializing in breast cancer evaluated the BC specimens, blind to patient outcomes for:

Pre- and post-NAC stromal TIL levels, following the recommendations of the international TILs Working Group [21,22] (mononuclear cell infiltrate, including lymphocytes and plasma cells, excluding polymorphonuclear leukocytes). TILs were studied in all patients, including those in pCR, in accordance with the latest guidelines. TILs were evaluated in the stroma, within the border of the tumor scar, after the exclusion of tumor areas with necrosis and artifacts, and were scored continuously as the mean percentage of the stromal area occupied by mononuclear cells.

Pathological complete response (pCR), defined as the absence of infiltrative carcinoma in the breast and lymph nodes. Persistent in situ carcinoma in the breast was considered a complete response [23].

Lymphovascular invasion (LVI), defined as the presence of carcinoma cells within a finite endothelium-lined area (a lymph or blood vessel), which was evaluated before and after NAC.

### 2.6. Study Endpoints

The objectives of the study were to analyze factors associated with pCR, relapse-free survival (RFS), and overall survival (OS). pCR was defined as the absence of an invasive residual tumor from both the breast and axillary nodes (ypT0/is N0). RFS was defined as the time from surgery to locoregional recurrence, distant metastasis or death, whichever occurred first, in patients without progressive disease or distant metastasis at the time of scheduled surgery. Overall survival (OS) was defined as the time from surgery to death. For patients for whom none of these events were recorded, data were censored at the time of last known contact. The cutoff date for survival analysis was 1 April 2019.

### 2.7. Statistical Analysis

The absolute change in TIL level was defined as the change in TIL levels during NAC. We distinguished between patients with decreasing TIL levels and those with constant or increasing TIL levels.

Prognostic models were generated and adapted according to the dependent variable: linear models for pre-NAC TIL levels, post-NAC TIL levels and the change in TIL levels, logistic models for pCR status, and Cox models for RFS and OS. For the pre- and post-NAC TIL level models, the logarithm of TIL level was used, to ensure that the assumption of normality was satisfied. In each case, univariate analyses were performed as a first step. Continuous variables were analyzed as such, and dichotomized according to their distribution. Variables significantly associated with the outcome at a 10% level were then included in a multivariate model. When both the continuous and dichotomized versions of a variable were selected, only the most significant was kept. Backward stepwise variable selection based on the Akaike information criterion (AIC) was also performed. Prognostic models for RFS were also constructed independently for the subcohort of patients not in pCR. An alpha risk of 5% was used for variable selection for the multivariate analysis. We handled the missing values for predictors by performing multiple imputation on the dataset. All the variables used in the analyses were included in the imputation model, including the outcomes (TIL levels, pCR status, OS and RFS event indicators, and Nelson-Aalen cumulative hazard rate estimators), as recommended in a previous study [24]. We generated 50 imputed datasets, and, according to the multiple imputation framework, all analyses were performed independently on each imputed dataset, with the results later pooled according to Rubin’s rules. The most frequent combination of selected variables over the 50 imputed datasets was used for the final multivariate model. When the absolute change in TIL levels was selected in addition to pre- or post-NAC levels, only the most frequently selected variable was retained, to prevent the duplication of information. For continuous variables, the log-linearity assumption was evaluated in univariate and multivariate models, by polynomial fit and deviance analysis. If this assumption was violated, the polynomial version of the variable was used. The proportional hazards assumption of Cox models was checked with Schoenfeld residuals. If this assumption was not valid, the predictor was not analyzed.

The distribution of pre-NAC TIL levels and post-NAC TIL levels according to different variables was displayed in boxplots, with the significance of differences between variable levels being assessed in univariate analyses, as described above. Correlations were assessed in two-tailed Pearson tests. Relapse-free and overall survival curves were plotted by the Kaplan–Meier method. All analyses were performed with R version 3.5.3. (Vienne, Austria).

## 3. Results

### 3.1. Baseline Characteristics of Patients and Tumors

We included 117 patients in the study (Figure 1). The baseline characteristics of these patients and their tumors are detailed in Table 1. The median age of the patients was 49.4 years. Most of the patients had large tumors, with 53% of tumors at the T3 stage and clinical nodal involvement observed in 61 patients (52.2%) (N1–N2–N3). In total, 98 patients (83.8%) had grade 3 tumors, and P53 was mutated in 77 patients (85.6%).

### 3.2. Metabolic on-Treatment Evaluation and Change in TIL Levels on NAC

The median pre-NAC TIL level was 15% and the mean level was 21% (Figure 2A,B). In univariate analysis (Appendix A), pre-NAC TIL level was not associated with any clinical or pathological parameter (Figure 2C,D) other than baseline SUV (*p* = 0.025, Figure 2E).

SUVmax was evaluated before NAC in 108 patients (92.3%) and after two cycles in 99 patients (84.3%). The median SUVmax value was 11.9 at baseline, falling to 3.8 after two cycles (Figure 3A). Forty-four patients (44%) displayed a large decrease in SUVmax after two courses of chemotherapy (more than 70% of the baseline value).

The factors associated with a decrease in SUV were a higher grade, a clinical tumor stage of T1 or T2 rather than T3, and high pre-NAC TIL levels (Appendix A). In multivariate analysis, tumor stage, tumor grade, and pre-NAC TIL levels were associated with a large decrease in SUV.

After chemotherapy, TIL levels had decreased in 69 patients (69%), had not changed in seven patients (7%), and had increased in 24 patients (24%). Overall, median post-NAC TIL levels were significantly lower after NAC than at baseline (3% versus 15% respectively, *p* < 0.0001) (Figure 3B). The change in TIL levels was significantly negatively associated with baseline SUV and change in SUV (rho = 0.44, *p* < 0.0001, Figure 3C), with pre-NAC TIL levels (Figure 3D) and with the occurrence of a pCR (Figure 3E, *p* = 0.003, Appendix A).

In the population of patients without a pCR, median pre-NAC and post-NAC TIL levels were 10% and 7%, respectively. After chemotherapy, TIL levels had decreased in 32 patients (47%), had not changed in six patients (8.8%) and had increased in 20 patients (29.4%). In the population of patients with a pCR, median pre-NAC and post-NAC TIL levels were 15% and 1%, respectively (Appendix A).

### 3.3. Response to Treatment and Post-NAC Tumor Characteristics

All but one of the patients in the SIM1 cohort had surgery after NAC, and this surgery was conservative in 74 patients (64.3%). A pathological complete response (pCR) was achieved in 49 (41.8%) patients. Post-NAC LVI was observed in 10 patients (8.6%). In univariate analysis, higher tumor grade, low post-NAC TIL levels, a decrease in SUV, and a decrease in TIL were significantly associated with pCR (Appendix A). Only low post-NAC TIL levels and the relative change in SUV remained significantly associated with pCR in multivariate analysis.

Post-NAC TIL levels were analyzed on surgery specimens from all but nine patients (with a lack of analysis mostly due to the absence of a scar in the surgery specimen). The median post-NAC TIL level was 3% (Figure 4A,B). High post-NAC TIL levels were significantly associated with a high SUV after two cycles (Figure 4C, Table 2), with an absence of pCR (Figure 4D, Table 2), and the presence of involved lymph nodes after NAC completion (Figure 4E, Table 2), and tended to be associated with the presence of post-NAC LVI (Figure 4F, Table 2).

### 3.4. Survival Analyses

#### 3.4.1. Relapse-Free Survival

With a median follow-up of 38 months, four patients presented a locoregional relapse and 23 had metastases. Three-year relapse-free survival was 79.1% (95% CI: (71.4%–87.5%)). In univariate analysis, the presence of both pre-NAC and post-NAC LVI, the absence of pCR, post-NAC nodal involvement, and higher post-NAC TIL levels (Figure 4G) were significantly associated with poor RFS (Table 3). In multivariate analysis, post-NAC LVI (HR = 3.1, 95% CI (1.2; 8.3), *p* = 0.02), and high post-NAC-TIL levels (HR = 1.4, 95% CI (1.1; 1.9), *p* = 0.014 per 10% increment of TIL levels) were independent predictors of poor RFS (Figure 4G, Table 3).

Similar results were obtained for an analysis focusing exclusively on the population of patients without pCR. Three-year relapse-free survival was 73.7% (95% CI: (62.9%–86.3%)). In multivariate analysis, post-NAC LVI (HR = 3.4, 95% CI (1.2; 9.3), *p* = 0.02), and high post-NAC-TIL levels (HR = 1.6, 95% CI (1.1; 2.1), *p* = 0.005 for an increment in TIL levels of 10 units) were independent predictors of poor RFS (Table 4).

#### 3.4.2. Overall Survival (OS)

In total, 16 patients died from breast cancer. Three-year overall survival was 88.7% (CI (82.5%; 95.2%)). In univariate Cox proportional hazards analyses, post-NAC LVI, post-NAC lymph node involvement, high post-NAC TIL levels (Figure 4H), and the changes in TIL levels and SUV were significantly associated with OS **(**Appendix A). In multivariate analysis, only post-NAC LVI (HR = 4.8, 95% CI (1.5; 15.7), *p* = 0.009) and high post-NAC TIL levels (HR = 1.8 (95% CI 1.3; 2.3), *p* < 0.0001, for an increment in TIL levels of 10 units) remained independent predictors of poor OS.

Similar results were obtained for the population without pCR. Post-NAC LVI (HR = 5.5, 95% CI (1.4; 22.1), *p* = 0.016) and high post-NAC TIL levels (HR = 2 (95% CI 1.4; 2.8), *p* = 0.0001, for an increment in TIL levels of 10 units were associated with poor survival in multivariate analysis (Appendix A).

## 4. Discussion

We report here analyses of pre- and post-NAC TIL levels in a cohort of TNBC patients treated by dd-di NAC. We found that high post-NAC TIL levels had an adverse effect on prognosis, increasing the risk of relapse and death. No previous study has ever investigated paired TNBC samples in a cohort of patients treated with a dose-dense chemotherapy regimen. Our results provide new insights for breast immuno-oncology in the neoadjuvant setting.

First, despite many studies reporting that pre-NAC TIL levels are predictive of the likelihood of achieving pCR in TNBC [4,5,7,25], we found no such association in our cohort. This lack of association may be due to the characteristics of our population, in which most of the tumors were large, with a high frequency of clinical node involvement, and low pre-NAC TIL levels. One large meta-analysis of 2148 patients with TNBC reported negative associations of pre-NAC TIL levels with age, tumor size, lymph node involvement, and histological grade [26]. Alternatively, the association of pre-NAC TIL levels with the response to chemotherapy may depend on the type of chemotherapy regimen, as previously reported in another large series [4,27]. Finally, given that the association between baseline TIL levels and pCR was close to statistical significance, the lack of association in our study may also result from a lack of statistical power.

Second, our results concerning the change in TIL levels are consistent with those of previous studies based on conventional TIL assessment [13] or digital pathology [28,29]. In all three studies, there was a significant negative association between the change in TIL levels and the likelihood of pCR. Our findings therefore add to the growing evidence that residual immune infiltration is significantly greater in tumors with a residual tumor burden than in specimens cleared of tumor cells by dd-di NAC.

Third, consistent with the finding of a significant association between higher post-NAC TIL levels and an absence of pCR, the presence of lymphovascular invasion, and a higher degree of nodal involvement, high post-NAC TIL levels had an independent adverse impact on oncological outcomes, in terms of both RFS and OS. This finding was unexpected, because three previous large cohort studies showed that high post-NAC TIL levels had either a positive effect [8,10], or no effect on prognosis [13]. However, none of the patients in these three cohorts was treated by dd-di NAC, and it is highly plausible that chemotherapy regimens and sequences have a considerable effect on the composition of the post-NAC immune tumor microenvironment. Previous studies suggested that TIL levels were related to pCR or disease-free survival, but in ways that differed between chemotherapy regimens. In the BIG 02-98 trial, Loi et al. found a significant interaction between increasing TIL levels and the benefits of anthracycline-only chemotherapy (*p* = 0.042) [27]. In the GeparSixto trial [4] pCR rates were higher in patients with lymphocytes predominant breast cancers treated with an anthracycline and taxane-based regimen to which carboplatin was added, but not in the group without carboplatin (*p* = 0.002 for interaction). Finally, in a retrospective study of 1318 of the 2994 patients included in the GAIN 1 study, TIL levels had a positive prognostic impact on DFS in the dose-dense epirubicin, paclitaxel and cyclophosphamide (EPC) arm, but not in the arm treated with this regimen plus capecitabine (*p* = 0.04) [30]. Overall, these results suggest that different CT regimens may have different effects on the predictive or prognostic impacts of TIL levels assessed in untreated tumors. Nevertheless, to our knowledge, this interaction (i.e., the dependence of the prognostic impact of post-NAC TILs on the chemotherapy regimen) has never been investigated and remains to be confirmed in large pooled analyses.

As these data are unprecedented, we therefore provide our original data as an open-access resource for the medical and scientific community, for pooling with existing datasets (Appendix A).

Persistently high TIL levels after chemotherapy may reveal (i) the onset of an imbalanced immune response, enriched in immunosuppressive infiltration; (ii) a global profile of chemoresistance (i.e., an inability of the chemotherapy to clear either tumor cells or immune infiltration).

Finally, we highlight the complexity of interpreting TIL levels and the importance of taking the NAC regimen into account when interpreting results. Tumors with high levels of TILs post-NAC were found to be at higher risk of relapse, suggesting that the addition of post-NAC immunotherapy might be beneficial in such patients. Despite an appealing translational rationale, this strategy remains to be confirmed in the setting of dedicated adjuvant post-NAC immunotherapy trials.

## Figures and Tables

**Figure 1 cancers-14-01331-f001:**
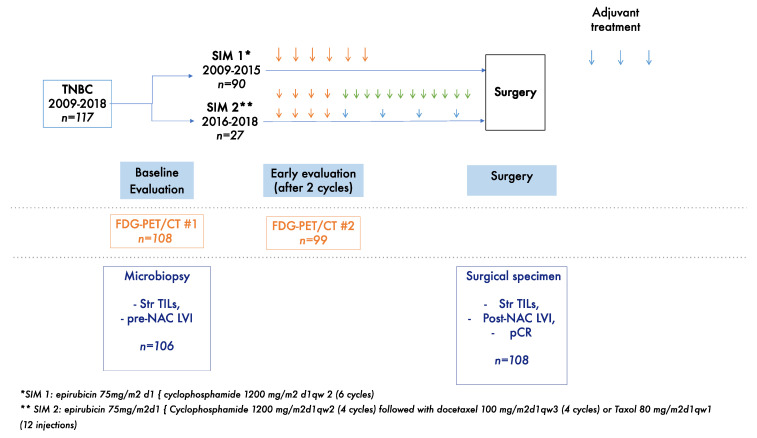
Study diagram. Orange arrows represent dose dense dose intense anthracycline and cyclophosphamide-based chemotherapy; blue arrows represent taxanes based chemotherapy; green arrows represent weekly paclitaxel chemotherapy.

**Figure 2 cancers-14-01331-f002:**
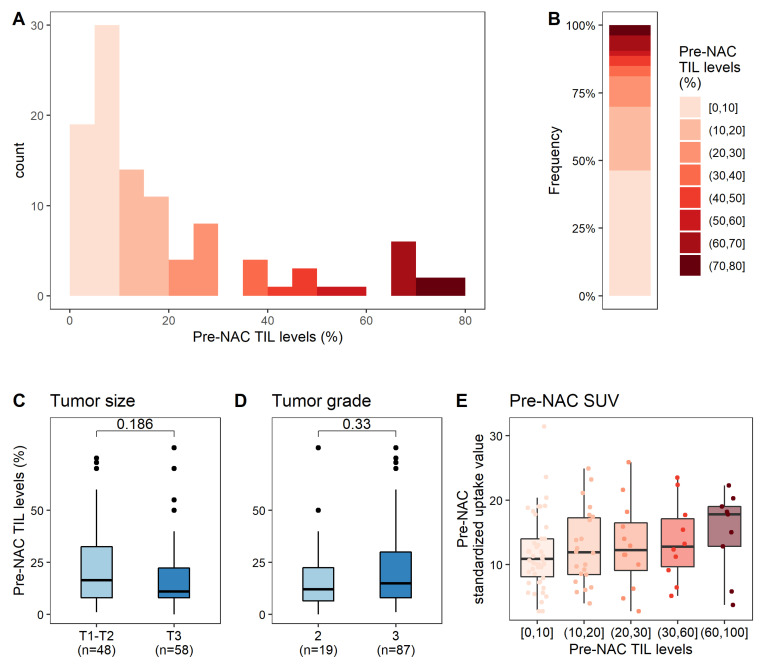
Associations between pre-NAC TIL levels and clinical, pathological, and metabolic factors. (**A**) Distribution of pre-NAC TIL levels; (**B**) Distribution of pre-NAC TIL levels, by increments of 10%; (**C**–**E**) Association of pre-NAC TIL levels with clinical and pathological factors: tumor size (**C**), tumor grade (**D**), pre-NAC SUV (**E**).

**Figure 3 cancers-14-01331-f003:**
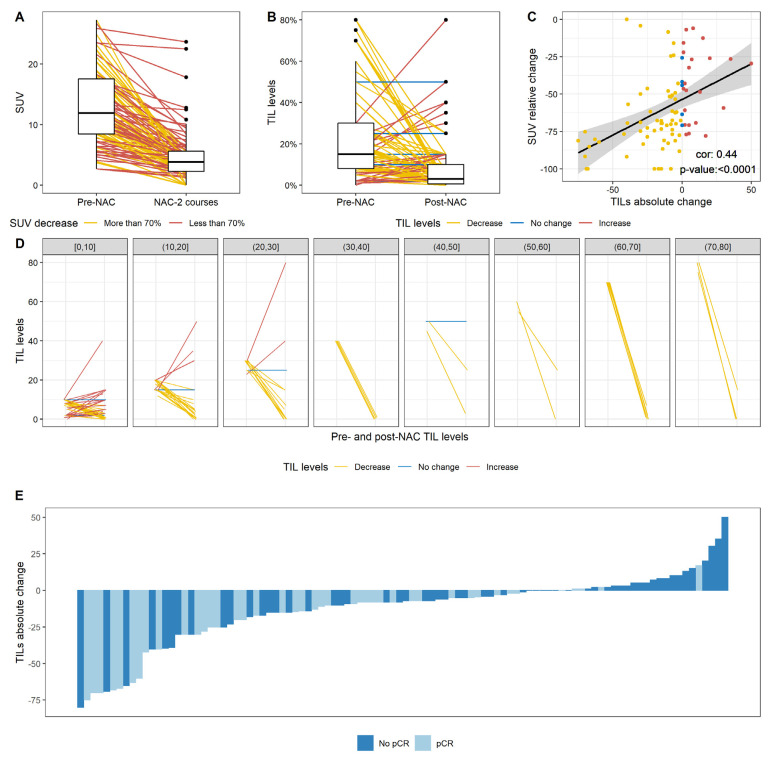
Change after NAC in SUV levels, TIL levels, and the correlation between the relative change in SUV and the absolute change in TIL levels. (**A**) Change in SUV levels after two courses of NAC. The lines indicate the pre NAC and post NAC (after two courses of NAC) paired SUV results for a given patient and are colored according to the category of change in SUV (more than 70%; less than 70%); (**B**) Change in TIL levels after NAC; (**C**) Pearson correlation between the relative change in SUV and the absolute change in TIL levels. Points are colored according to the category of change in TIL levels (decrease; no change; increase in TIL levels); (**D**) Change in TIL levels according to pre-NAC TIL level, binned by increments of 10%. (**B**,**D**) Lines indicate the paired pre and post NAC paired TIL levels for a given patient and are colored according to the category of change in TIL levels (decrease; no change; increase in TIL levels); (**E**) Waterfall plot representing the change in TIL levels according to pCR status; each bar represents one patient, and patients are ranked in ascending order of change in TIL levels.

**Figure 4 cancers-14-01331-f004:**
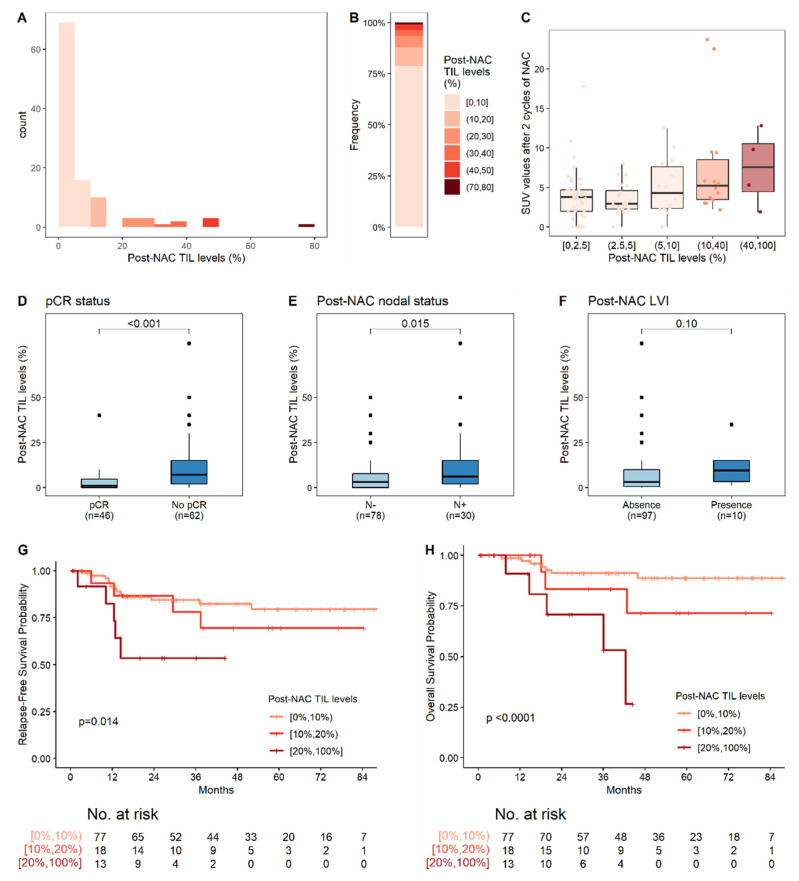
Post-NAC TIL levels and their association with post-NAC pathological factors. (**A**) Distribution of post-NAC TIL levels; (**B**) Distribution of post-NAC TIL levels, by increments of 10%; (**C**) Association of post-NAC TIL levels with SUV values after two cycles of NAC; (**D**–**F**) Association of post-NAC TIL levels with clinical and pathological factors: pCR status (**D**); post-NAC nodal involvement (**E**); and post-NAC LVI (**F**); (**G**,**H**) RFS (**G**) and OS (**H**) according to post-NAC TIL level, stratified into three classes: 0–10%; 10–20%; ≥20%. The reported *p*-values are those for post-NAC TIL levels in the corresponding multivariate Cox regression (i.e., the continuous TIL levels, after adjustment for post-NAC nodal involvement and post-NAC LVI for RFS (**G**) and for post-NAC LVI for OS (**H**)). The corresponding Cox regressions are reported in Table 3 and Appendix A, respectively.

**Table 1 cancers-14-01331-t001:** Patients and tumor characteristics.

Parameters	Values	*n* = 117	Statistics *
**Pre-NAC parameters**			
**Age**		117	49.37 (23.69; 72)
**Pregnancies**	0	33	28.2%
	1–3	66	56.4%
	>3	18	15.4%
**Menopausal status**	premenopausal	68	58.1%
	postmenopausal	49	41.9%
**Family history**	0	82	71.3%
	1	21	18.3%
	≥2	12	10.4%
**Clinical tumor stage**	T1	3	2.5%
	T2	52	44.00%
	T3	62	53%
**Clinical nodal status**	N−	56	47.9%
	N+	61	52.1%
**Tumor grade**	2	19	16.2%
	3	98	83.8%
**p53 (FASAY)**	Mutated	77	85.6%
	Wild type	13	14.4%
**LVI**	Absence	91	83.5%
	Presence	18	16.5%
**SUV**		108	11.95 (2.7; 31.4)
**TIL levels**		106	15 (0; 80)
**Chemotherapy regimen**	SIM-1	90	76.9%
	SIM-2	27	23.1%
**Post-NAC parameters**			
**pCR**	No pCR	68	58.1%
	pCR	49	41.9%
**Nodal involvment**	ypN−	83	71.5%
	ypN+	33	28.4%
**LVI**	Absence	106	91.4%
	Presence	10	8.6%
**SUV at 2 courses**		99	3.8 (0; 23.7)
**TIL levels**		108	3 (0; 80)
**Variation Pre/Post-NAC parameters**			
**SUV relative variation**		100	−68.77 (−100; 0)
**TILs absolute variation**	<0	69	69%

* percentage for class parameters; median (range) for continuous parameters. Number of unavailable information: Family history (2); p53 (27); Pre-NAC LVI (8); Pre-NAC SUV (9); Pre-NAC TIL levels (11); Post-NAC nodal involvement (1) Post-NAC LVI (1); SUV at 2-cures (18); Post-NAC TIL levels (9); SUV relative variation (17); TILs absolute variation (17).

**Table 2 cancers-14-01331-t002:** Predictive factors associated with post-NAC TIL levels.

		Univariate Analyses			Multivariate Analysis		
Parameters	Levels	Coefficient	IC 95%	*p*-Value	Coefficient	IC 95%	*p*-Value
**Pre-NAC parameters**							
Age		−0.01	(−0.03; 0.0097)	0.31			
Age (class)	<45						
	≥45	−0.089	(−0.59; 0.41)	0.726			
Pregnancies	0						
	1–3	0.17	(−0.36; 0.7)	0.534			
	>3	0.0098	(−0.71; 0.73)	0.978			
Menopausal status	premenopausal						
	postmenopausal	−0.13	(−0.59; 0.34)	0.595			
Family history	0						
	1	0.15	(−0.45; 0.75)	0.63			
	≥2	0.53	(−0.21; 1.3)	0.16			
**Clinical tumor stage**	**T1–T2**						
	**T3**	**0.45**	**(−0.000091; 0.9)**	**0.05**			
Clinical nodal status	N−						
	N+	0.16	(−0.3; 0.61)	0.5			
Tumor grade	2						
	3	0.06	(−0.56; 0.68)	0.847			
LVI	Absence						
	Presence	0.11	(−0.54; 0.75)	0.74			
SUV		−0.026	(−0.065; 0.014)	0.20			
TIL levels		−0.0027	(−0.014; 0.0083)	0.63			
Chemotherapy regimen	SIM-1						
	SIM-2	0.43	(−0.12; 0.98)	0.127			
**Post-NAC parameters**							
**pCR**	**No pCR**						
	**pCR**	**−1.1**	**(−1.5; −0.69)**	**<0.001 *****	**−0.89**	**(−1.4; −0.38)**	**0.0008 *****

**Nodal involvment**	**ypN−**						
	**ypN+**	**0.62**	**(0.12; 1.1)**	**0.015 ***			
LVI	Absence						
	Presence	0.65	(−0.13; 1.4)	0.10			
**SUV at 2 cures**		**0.072**	**(0.012; 0.13)**	**0.020 ***			
**Variation Pre/Post-NAC parameters**							
**SUV relative variation**		**0.017**	**(0.0083; 0.025)**	**0.0002 *****	0.0075	(−0.0026; 0.018)	0.14
**SUV relative variation (class)**	**<−70%**						
	**≥−70%**	**0.83**	**(0.37; 1.3)**	**0.0005 *****			

*p*-Values were annotated as follow: *: ≤0.05; ***: ≤0.001.

**Table 3 cancers-14-01331-t003:** Predictive factors associated with relapse-free survival in the whole population.

		Univariate Analyses			Multivariate Analysis		
Parameters	Levels	HR	(95% CI)	*p*-Value	HR	(95% CI)	*p*-Value
**Pre-NAC parameters**							
Age (class)	<45	1					
	≥45	0.67	(0.29; 1.5)	0.34			
Pregnancies	0	1					
	1–3	0.69	(0.29; 1.6)	0.41			
	>3	0.65	(0.18; 2.4)	0.52			
Menopausal status	premenopausal	1					
	postmenopausal	0.57	(0.24; 1.4)	0.21			
Family history	0	1					
	1	1	(0.34; 3)	0.98			
	≥2	1.5	(0.44; 5.2)	0.50			
Clinical tumor stage	T1-T2	1					
	T3	1.2	(0.52; 2.6)	0.71			
Clinical nodal status	N−	1					
	N+	1.9	(0.82; 4.5)	0.13			
Tumor grade	2	1					
	3	4	(0.53; 29.3)	0.18			
**LVI**	**Absence**	**1**					
	**Presence**	**3.1**	**(1.3; 7.5)**	**0.010 ***			
TIL levels (continuous) ^a^		0.99	(0.97; 1)	0.43			
**Post-NAC parameters**							
**pCR**	**No pCR**	**1**					
	**pCR**	**0.39**	**(0.16; 0.99)**	**0.047 ***			
**Nodal involvment**	**ypN−**	**1**			1		
	**ypN+**	**3.3**	**(1.5; 7.3)**	**0.004 ****	2.2	(0.91; 5.2)	0.080
**LVI**	**Absence**	**1**			**1**		
	**Presence**	**4.6**	**(1.8; 11.6)**	**0.001 ****	**3.1**	**(1.2; 8.3)**	**0.020 ***
**TIL levels (continuous) ^b^**		**1.5**	**(1.2; 2)**	**0.002 ****	**1.4**	**(1.1; 1.9)**	**0.014 ***
**Variation Pre/Post-NAC parameters**							
SUV relative variation ^c^				0.11			
SUV relative variation (class)	<−70%	1					
	≥−70%	1.3	(0.55; 2.9)	0.59			
TILs absolute variation	<0	1					
	≥0	1.9	(0.81; 4.5)	0.14			

^a^ Cox proportional risk hypothesis not verified as continuous variable. With the variable binned with a 20% cut-off, coefficients are as follows: HR (≥20% versus <20%, reference class) = 0.74, 95%CI (0.29; 1.8), *p* = 0.51. ^b^ this HR corresponds to difference of 10 units of the variable (for example, the univariate analysis reveals that a patient with post-NAC TIL levels of 25% will have 1.5 times the risk of relapse or death than that of a patient with a level of 15%). ^c^ continuous variable modelized with a degree 3 polynomial (coefficients and IC: 0.85 (0.7; 1); 1 (0.99; 1); 1 (1; 1)). *p*-Values were annotated as follow: *: ≤0.05; **: ≤0.01.

**Table 4 cancers-14-01331-t004:** Predictive factors associated with RFS for the Non-pCR subcohort.

		Univariate Analyses			Multivariate Analysis		
Parameters	Levels	HR	(95% CI)	*p*-Value	HR	(95% CI)	*p*-Value
**Pre-NAC parameters**							
Age (class)	<45	1					
	≥45	0.9	(0.34; 2.4)	0.84			
Pregnancies	0	1					
	1–3	1.3	(0.45; 3.7)	0.63			
	>3	0.81	(0.16; 4.2)	0.80			
Menopausal status	premenopausal	1					
	postmenopausal	0.46	(0.16; 1.3)	0.14			
Family history	0	1					
	1	0.58	(0.13; 2.6)	0.47			
	≥2	1	(0.23; 4.4)	1			
Clinical tumor stage	T1-T2	1					
	T3	0.91	(0.36; 2.3)	0.85			
Clinical nodal status	N−	1					
	N+	1.4	(0.54; 3.6)	0.50			
LVI	Absence	1					
	Presence	2.3	(0.8; 6.4)	0.12			
TIL levels (continuous)		1	(0.99; 1)	0.45			
**Post-NAC parameters**							
Nodal involvment	ypN−	1			1		
	ypN+	2.6	(0.94; 7.4)	0.066	2.2	(0.76; 6.4)	0.15
**LVI**	**Absence**	**1**			**1**		
	**Presence**	**3.6**	**(1.3; 9.5)**	**0.012 ***	**3.4**	**(1.2; 9.3)**	**0.020 ***
**TIL levels (continuous) ^a^**		**1.6**	**(1.1; 2.1)**	**0.005 ****	**1.6**	**(1.1; 2.1)**	**0.005 ****
**Variation Pre/Post-NAC parameters**							
SUV relative variation		0.99	(0.97; 1)	0.17			
SUV relative variation (class)	<−70%	1					
	≥−70%	0.65	(0.22; 1.9)	0.44			
TILs absolute variation	<0	1					
	≥0	1.8	(0.67; 4.9)	0.24			

^a^ This HR corresponds to difference of 10 units of the variable (for example, the univariate analysis reveals that a patient with post-NAC TIL levels of 20% will have 1.6 times the risk of relapse or death than that of a patient with a level of 10%). *p*-Values were annotated as follow: *: ≤0.05; **: ≤0.01.

## Data Availability

The authors provide fully deidentified metadata as Appendix A, together with R scripts, that have been added as Appendix A.

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
