# Peer review of "Negative Relationship between Post-Treatment Stromal Tumor-Infiltrating Lymphocyte (TIL) and Survival in Triple-Negative Breast Cancer Patients Treated with Dose-Dense Dose-Intense NeoAdjuvant Chemotherapy"

_cancers, 2022, doi:10.3390/cancers14051331_

Round 1

Reviewer 1 Report

Summary

This is a well-written manuscript by Giacchetti et al. The authors retrospectively investigated 117 TNBC tumors treated with dose-dense and dose intense (dd-di) neoadjuvant chemotherapy at the Breast Disease Center of Saint Louis University Hospital in Paris between 2009 and 2018. Most patients had paired pre-NAC and post-NAC information which is unique and useful. The authors found that, unlike many previous reports, pre-NAC TIIL levels in this cohort did not show a significant association with pCR. The discussion of this difference from the authors are reasonable, although further investigation is required to draw a conclusion. The author further revealed a negative association between the change in TIL levels and pCR, consistent with previous studies. However, the authors found a negative association between high post-NAC TIL and low pCR rate and worst prognosis, which is the opposite from many previous studies of large cohorts. The unique dd-di treatment regimens and sequences may explain the difference in this observation.

Overall, I find this study of major clinical significance in terms of 1) interpreting TIL levels for prognosis; 2) guiding potential novel treatment strategies

Major:

None

Minor:

  1. Figure 1, the year label in TNBC and its two subgroups are inconsistent. TNBC is 2011-2017, while SIM1* subgroup is 2010-2015. In the materials and methods section, it is from Jan 2009. These statements should be consistent.
  2. Suggest putting components in figure 1 in different rows with edges, so that the relationship of each layer of information is more straightforward to follow. Adding a detailed figure legend for the figure1 could also help make it more clear.
  3. authors should provide the metadata table for each sample. This allows further exploration of the data and validation of the results by others using different methods. Currently only summarized characteristics are included in table 1.

Reviewer 2 Report

The manuscript provides new data about TIL during dose-dense dose-intense Neoadjuvant chemotherapy (dd-di-NAC).  While multiple studies evaluated TIL before (and occasionally after) conventional NAC, little is known about TIL during dd-di-NAC.  So, this manuscript could be of interest for researchers and clinicians optimising dd-ds-NAC in TNBC. 

The following points should be corrected/addressed before the manuscript can be published:

1) Reporting TIL levels should be made consistent and clear

In the current version of the manuscript, TIL levels are reported in multiple different ways.  For instance, pre-treatment section in Table 4 reports TIL as a continuous variable and as a factor with cut-off at 20%; then the same table reports TIL post-NAC as continuous variable with 10% increment and as a factor with cut-off at 10%.  Supplementary Table 3 reports pre-treatment TIL levels as “x” and “x-squared”.  Supplementary Table 2 reports TIL as “x”, “x-squared” and “x-cubed”.  It’s confusing.  

It would make the results presentation clear, if all tables reported TIL consistently e.g. as a continuous variable, possibly with 10% increment (or another consistent way that the authors prefer). 

2) Clarify x and “x squared” notations in tables, or remove this notation when it is not necessary

Not only TIL is occasionally reported in this way.  For instance, Table 4 reports age (!) as “x”, “x-squared” and “x-cubed”.  This notation should be either explained or changed to something more conventional (e.g. why not to analyse age as a continuous variable expressed in years?). 

1) Add some information about clinical responses

Some information about clinical response could be added to section 3.3 (Response to treatment and post-NAC tumor Characteristics).  Also, paragraph starting on line 137 describes treatment of patients when no tumour progression was observed.  What happened if the tumour progressed on treatment?  

Minor points, which still should be corrected prior publication:

4) Correct errors in referencing

e.g.:

Line 67: Check reference 16 (should it be changed to 17?)

Line 157: Check reference 21 (should it be changed to 24?)

Line 162: Check reference 21 (should it be changed to 25?)

Reference 13 in the list of references: reformat consistently with other references (check detail in doi 10.1158/1078-0432.CCR-18-3017 )

5) Correct potentially misleading wording

Lines 326-327

The manuscript can be read as “meta-analysis … reported negative associations of pre-NAC TIL levels with … immune infiltration”.  It should be rephrased (TIL is the immune infiltration, it can’t be negatively associated with itself). 

Lines 366-367

The manuscript says: “Our study opens up new possibilities for treatment”.  It is easy to misinterpret this as an overstatement.  In fact, the study did not introduce any new treatments, it only evaluated biomarkers.  This sentence should be rephrased or removed. 

The remaining points could be left on the authors’ discretion:

6) Line 28 in Abstract

The abstract says: “Few studies have analyzed changes in TIL levels following NAC”.  It might be more relevant to emphasise “following di-dd-NAC” here. 

7) Section 2.7 Statistical analysis

Although it is not yet a standard practice in the studies of TIL, the authors might consider adding supplementary files with key elements of R code and with primary data.  This would significantly improve the report, considering current trend toward reproducible research.  

8) Line 214

Words “locally advanced tumours” might be changed to “large tumours” to avoid confusion with locally advanced cancer involving skin or chest. 

9) Table 1

Why to report Androgen receptors?  They do not look relevant to this paper.  If you keep them, then correct the typo in the table notes (RA should be AR). 

Why to include chemotherapy protocols in the table notes?  You may just refer to section 2.3 of Methods and to Figure 1.

10) Line 350

LPBC abbreviation may need to be explained. 

11) Line 358

Words “this interact” may be changed to “this interaction”.
